# Diagnostic Concordance between the Visual Analogue Anxiety Scale (VAS-A) and the State-Trait Anxiety Inventory (STAI) in Nursing Students during the COVID-19 Pandemic

**DOI:** 10.3390/ijerph19127053

**Published:** 2022-06-09

**Authors:** Ana Lavedán Santamaría, Olga Masot, Olga Canet Velez, Teresa Botigué, Tània Cemeli Sánchez, Judith Roca

**Affiliations:** 1Department of Nursing and Physiotherapy, University of Lleida, 25003 Lleida, Spain; ana.lavedan@udl.cat (A.L.S.); teresa.botigue@udl.cat (T.B.); tania.cemeli@udl.cat (T.C.S.); judith.roca@udl.cat (J.R.); 2Health Care Research Group (GRECS), Biomedical Research Institute of Lleida, 25003 Lleida, Spain; 3Faculty of Health Sciences Blanquerna, University Ramon Llull, 08022 Barcelona, Spain; olgacv@blanquerna.url.edu

**Keywords:** anxiety, COVID-19, nursing students, state-trait anxiety inventory, visual analogue scale

## Abstract

Anxiety is one of the most common problems among nursing students. The State-Trait Anxiety Inventory (STAI) is used to detect anxiety in this population; however, its length hinders speedy detection. For this reason, a faster and more efficient instrument is needed for early detection. This study aimed to determine the relationship between the anxiety measurement scales State-Trait Anxiety Inventory (STAI) and the Visual Analogue Scale for Anxiety (VAS-A) by establishing a discrimination threshold through the contrast of true positive rates (VPR) and false positive rates (FPR). To this end, a cross-sectional quantitative observational and analytical study was carried out on 185 fourth-year nursing students. The data collected were anxiety (STAI and VAS-A) and socio-demographic variables during the COVID-19 pandemic. The results showed a correlation between the two scales (VAS-A and STAI). The VAS-A is a useful instrument for assessing students in a crisis that could potentially generate anxiety. The study established a reasonably safe error probability range (>5%), allowing the VAS-A scale to be used as a rapid diagnostic or pre-diagnostic tool, depending on the scores. The study shows that speedy detection of anxiety using the VAS-A and an in-depth approach with the STAI by teaching staff in crises is possible.

## 1. Introduction

Anxiety is one of the most prevalent problems among university students [1]. Nursing students face several challenges throughout their education that can significantly impact their well-being [2], making them a special group. Simpson and Sawatzky, (2020) [3] define anxiety as the cognitive perception of a potential vague or ambiguous, subjective threat, producing psychological and physiological responses and behavioral changes. Nursing students experience much higher levels of anxiety than other health-related professionals and the general population [4,5]. Nursing students not only have academic, social, and personal challenges relating to their university studies. They also have to cope with the additional demands associated with clinical practice [6] or recently implemented virtual classes [7].

Moreover, the last two years have seen an unprecedented change in university teaching due to COVID-19. The current pandemic situation of COVID-19 has had a significant impact on nursing students’ emotional well-being, not least their level of anxiety. Traumatic events and prolonged home quarantine during epidemic periods increase the likelihood of anxiety [8]. In this regard, a recent study [9] comparing the mental health of final-year nursing degree students from different Spanish universities before and after the pandemic found that the pandemic compromised their mental well-being. The Covid-19 pandemic also caused constant disruptions and led to changes in nursing education programs [10]. For example, virtual or hybrid learning environments were introduced [11,12], and clinical placements were suspended and replaced by other models, such as the “auxiliary health staff” initiative [13,14,15], which caused more uncertainty and, consequently, greater anxiety in students. Hence, such anxiety affects learning and contributes to a decline in academic performance [16].

Thus, educational institutions require policies and procedures to protect students in health crises and, as a part of these protocols, instruments to detect and assess anxiety or stress [17]. In unprecedented circumstances in which, according to the evidence, students are already highly susceptible, there is a need for measurement instruments that are efficient, fast, and easy to use in health crises, which can generate anxiety in students. Several tools can be used to detect anxiety in this population. One of the most widely used by nursing students in academic and clinical settings [3] is the State-Trait Anxiety Inventory (STAI) [18,19]. However, depending on the setting, the STAI may be too long and too difficult to complete [20] since it has 20 items for assessing trait anxiety and 20 for state anxiety with polytomous items. Moreover, nurse academics’ heavy workloads make the tool too laborious to administer [21]. Indeed, there is a growing need for shorter measures that are nevertheless reliable. Previous attempts to create a short and valid version of the STAI had methodological shortcomings that limited the potential of the abbreviated forms as reliable tools for assessing anxiety in research and clinical practice or with a study population of the elderly [20,22].

Hence, there is a need for simple and short instruments that can be quickly self-administered individually in groups or on a mass scale. Such a tool would be extremely useful in acute crises. Therefore, a flexible instrument allowing the early detection of anxiety in nursing students is needed for undertaking a subsequent in-depth analysis with the STAI, if deemed necessary. Although the visual analogue scale of anxiety (VAS-A) provides a rapid (measurable and reproducible) classification of symptom severity [23], there are insufficient statistical data in the academic setting on the validity and reliability of this instrument for use in place of the STAI. In addition, to make an informed decision about which instruments should be administered, it is imperative to evaluate and report on these measures [5].

The lack of comparative studies comparing the two scales in assessing anxiety in nursing students led to the two-fold aim of this study: to determine the relationship between the anxiety scales (STAI and VAS-A) by establishing a discrimination threshold using the true positive rate (VPR) versus the false-positive rate (FPR) to suggest protocols for a more accurate diagnosis.

## 2. Materials and Methods

### 2.1. Design and Participants

This was a cross-sectional, quantitative observational, and analytical study. The study population consisted of nursing students from each of the four academic years at the Faculty of Nursing and Physiotherapy at the University of Lleida (Spain). The Nursing Degree in Spain has a duration of 4 years, totaling 240 European Credit Transfer and Accumulation System (ECTS), equivalent to 6000 h of theoretical and practical training. Approximately 360 students from all years were asked to voluntarily participate in the study. The sample size was calculated according to proportion (confidence level 99%, precision 3% and p-ratio = 0.5), giving a minimum value of 178; however, the final sample was larger than the estimated 185 students.

### 2.2. Measuring Instruments

The socio-demographic variables included age and sex, academic year, and healthcare worker.

Anxiety was assessed using two instruments. The students were asked to first indicate their anxiety level at a given moment using the VAS-A [24], a 10-cm horizontal line (from 0 to 10 points), which measured from left to right, absence or less intensity of anxiety to highest intensity of anxiety. A positive score was observed as equal to or higher than 6 points [25].

The second instrument used was the STAI [18,19], which has been shown to be useful in university settings [26]. The STAI features two subscales to measure two independent concepts of anxiety: State Anxiety Scale (S-Anxiety) and Trait Anxiety Scale (T-Anxiety). State anxiety is defined as an organism’s transient emotional condition, characterized by subjective feelings of tension and apprehension [18]. Trait anxiety is understood as a stable, anxious propensity to perceive people and situations as threatening, thus raising anxiety. Both subscales have 20 items scored on a Likert-type scale, with four response options (0 to 3). The questionnaire version that has been used previously in different Spanish samples shows good internal consistency [27], specifically among university students [26]. A result was considered positive when a score equal to or higher than 30 points was obtained in each subscale, which, in the case of state anxiety, corresponds to P85 in men and P90 in women, and in trait anxiety, to P85 and P77, respectively [18].

### 2.3. Data Collection and Ethical Considerations

Data were collected through an online questionnaire using the Google Forms application, including the variables under study and the measurement scales administered during the first outbreak of the COVID-19 pandemic in Spain (spring 2020). A link to the questionnaire was emailed to the participants via the Sakai^®^ platform. Participation was voluntary, and no compensation was offered.

After freely accessing the questionnaire, the students read an explanation of the purpose and nature of the study. They were also informed that each questionnaire would be assigned an alphanumeric code, guaranteeing data confidentiality and anonymity throughout the process. Before responding to the questionnaire, the participating students gave their informed consent. Prior to data collection, the project received the approval of the Study Committee of the Faculty of Nursing and Physiotherapy as the competent body of the university.

### 2.4. Data Analysis

First, to perform a descriptive analysis of the sample and identify variables associated with anxiety, descriptive and bivariate analyses were carried out according to the nature of the variables (X2, T-student, or test-Z). Additionally, to analyze the effect size, Hedge’s and Cohen’s tests were done.

Second, to achieve the aim of the study, the two scales were first tested for normality (Kolmogorov–Smirnov), reliability (Cronbach’s alpha), and correlation (Cronbach’s alpha). The VAS-A test was then compared with that of the S-Anxiety for diagnostic accuracy by referencing the receiver operating characteristic (ROC) curves and their related area under the curve (AUC), with confidence intervals of 95% CI. Finally, cut-off points were established to indicate anxiety (VAS-A score > 30 and S-Anxiety score > 6) by locating the thresholds where the highest sensitivity and specificity were reached and the zones of uncertainty. Thus, the discriminative power of the VAS-A test was estimated through error and hit probability to differentiate between people affected or not affected by anxiety.

The data were analyzed using IBM SPSS Statistics 24, and a significance level was set at *p* < 0.05 for all analyses.

## 3. Results

The sample consisted of 185 students, 88.1% of whom were women, with a mean age of 21.3 years (±4.0). The level of anxiety recorded by the VAS-A showed a mean of 6.19 out of 10 points (±2.09), and the STAI recorded a level of 58.43 out of 120 points (±18.81). Table 1 shows the rest of the sample characteristics.

The VAS-A scale and the S-Anxiety subscale measure the current state of anxiety, not a permanent trait. Table 2 shows the variables that were significantly associated with anxiety. As shown, academic year (*p* < 0.001 and *p* = 0.27, respectively) was the only variable associated with both the VAS-A and the S-Anxiety, with a medium effect size in VAS-A (Hedge’s *g* = 0.68) and very small in S-Anxiety (Hedge’s *g* = 0.07).

Concordance between the anxiety scales (STAI and VAS-A) used to measure anxiety (STAI and VAS-A) and the discrimination threshold was determined by calculating the sum of the variables of both STAI subscales (S-Anxiety and T-Anxiety). Normality tests (Kolmogorov–Smirnov) were carried out, showing a normal distribution for both S-Anxiety (KS = 0.055) and T-Anxiety (KS = 0.054). The reliability analysis of the total STAI instrument yielded an internal consistency α = 0.941, as did its two subscales, S-Anxiety (α = 0.929) and T-Anxiety (α = 0.887). In the question deletion tests, no item significantly altered Cronbach’s alpha.

The application of linear correlation tests between the continuous scales and the subscales (Table 3) highlighted a high positive correlation between the VAS-A and the S-Anxiety (r = 0.686; *p* = 0.000) and a low correlation between the VAS-A and the T-Anxiety (r = 0.417; *p* = 0.000).

To understand the interaction of the VAS-A and S-Anxiety scales in the same sample, the sums of the variables were dichotomized (No/Yes) by setting cut-off points at values >6 for VAS and >30 for S-Anxiety, following the literature [18,25]. Figure 1 shows the distribution of the two scales.

Although the figures appear similar, the diagnostic coincidence is 75%. There were overlaps of 46 mismatches between the VAS-A scale and the S-Anxiety (STAI). Table 4 shows the overlapping values. Of the 185 cases, VAS-A classified 32 cases as positives and S-Anxiety as negatives, and VAS-A classified a further 14 cases as negatives and S-Anxiety as positives.

The analysis for ROC allowed us to determine the diagnostic accuracy of the VAS-A scale by comparing it with the S-Anxiety scale to locate the cut-off points where greater sensitivity and specificity were achieved. This also allowed us to compare the discriminative capacity of the different cut-off levels of the VAS-A scale, that is, its ability to differentiate between healthy and sick subjects.

The AUC of the ROC was calculated for VAS-A (Figure 2), with S-Anxiety cut-offs equal to and higher than 30 points, confirming the values of ROC (0.816), confidence interval (0.753–0.879), and standard error (0.032, *p* = 0.000). ROC values >0.70 are considered robust.

Following the statistical analysis of the ROC coordinates, the smallest cut-off value observed was the minimum test value minus 1 (1–1), and the highest cut-off value was the maximum plus 1 (10 + 1). Thus, the point scores for each VAS-A cut-off value were estimated using the averages of the two consecutive test values observed. Table 5 shows the probability of error of the higher VAS-A (−) Negative coordinates calculated by Sensitivity, since there were no positives for scores below 6 points. The probability of finding false negatives was then inverted, while in the lower VAS-A (+) Positive coordinates, the probability of 1—Specificity was maintained.

As shown, when the VAS-A values are between 1 and 4, there is less than a 5% error in a false negative compared with the S-Anxiety. Similarly, when the VAS-A score is between 8 and 10 points, a false positive error is also lower than 5%. Only values between 5 and 7 in VAS-A have differences ranging from 10% to 33% error.

## 4. Discussion

This study presents the results of the concordance analysis between the VAS-A scale and the S-Anxiety (STAI) in undergraduate nursing students. For this purpose, a proportional sample of students from the different years of the four-year degree program was analyzed. The mean age of the participants, who were primarily women, was 21 years (SD: 4.0). Unlike studies conducted before the COVID-19 pandemic [28,29], this research showed greater levels of anxiety with higher means (Table 1). Furthermore, the results indicate that first- and second-year nursing students are at higher risk of anxiety than students in their final years (*p* = 0.027). In addition, depending on the effect size, the VAS-A shows more sensitivity (Hedge’s g = 0.68 vs. 0.07 in S-Anxiety) to the difference between groups of each nursing degree year. This finding conflicts with previous studies where higher anxiety scores were detected in students in higher years with more experience and a higher clinical workload [2,30]. Thus, it is reasonable to assume that as students mature with age and experience, they can better manage their emotions. Training and experience are two key elements of adaptation [13].

A positive correlation was found between the VAS-A and the S-Anxiety (0.686), thus the simplicity and ease of data collection of the VAS-A can be a useful instrument, by itself, for assessing anxiety in nursing students in crises such as the COVID-19 pandemic. Furthermore, AUC/ROC values >0.70 (0.816) in areas such as applied psychology and behavioral prediction are considered robust [31]. Thus, the ranges of concordance between the scales allow the instrument to be used as a diagnostic scale in the lowest scores (from 1 to 4) or the highest (from 8 to 10), with a margin of error of 0% to 4.2%. Due to the variations in error found (from 10% to 33%) in scores of 5, 6, and 7, the VAS-A should be administered together with the S-Anxiety. Scores between 5 and 7 accounted for 49% of the cases. Thus, the use of the VAS-A scale would significantly reduce the need to apply the S-Anxiety since a 51% reduction in its use in similar crises or emergencies is estimated. According to these results, the VAS-A could constitute a useful instrument for consecutive, longitudinal, control, and follow-up assessments in ongoing crises, based on detecting fluctuations between the lowest and highest scores. Our results support that the VAS-A used proactively can then set up interventions to reduce nursing anxiety to foster a more stable, adaptive, and meaningful nursing students’ group and teaching team to be better equipped for challenges that may not be easily predicated, such as the pandemic. To date, no previous research has been found that corroborates the results of our research in this population group (nursing students or other groups of university students) or in the academic context. However, similar results in different contexts were reported by authors including Lesage, Berjot, and Deschamps [32], who tested the VAS-A during the medical consultation, where a practitioner assessed the stress of all participants. They support our finding that VAS instruments are efficient and straightforward. In other contexts, this tool could also be used to parse out populations. For example, detecting the more emotionally stable and less emotionally stable health personnel could be very useful to address health demands with greater quality.

These instruments are also discriminatory in actually measuring what they are designed to assess. Multiple clinical studies [33,34] corroborate the use of these scales and show a correlation between the VAS-A and the STAI.

This research took the scores of the STAI adaptation proposed by Fonseca-Pedrero et al. [26] as its reference, setting the cut-off value at >30. However, this proposal should be reviewed in other studies and, if necessary, adapted to the context. Perception instruments vary over time. The mean scores in the STAI adaptation varied significantly, taking the first adaptation as a reference [27]. For this reason, some authors [33] who have evaluated the S-Anxiety cut-off at values >40 suggest that this is a debatable parameter. This could be explained by the loss of sensitivity of some anxiety assessment items over time and by the results of studies that have reviewed and validated the scales.

Other questions concerned the administration of the scales. In the present study, the scales were self-administered online. Therefore, as Abend, Dan, Maoz, Raz, and Bar-Haim (2014) [35] have highlighted, they can be administered simply and quickly, constituting a beneficial self-assessment instrument for large groups or in circumstances where presentiality is not possible, unnecessary, or paper-and-pencil questionnaires are impractical. In addition, the response order of the scales was also taken into account. According to the proposal of Labaste et al. [33], the initial use of the STAI could affect the mood of the participant/assessed and influence the response of the VAS-A; therefore, the students responded first with the VAS-A, followed then with the STAI.

### Limitations

The main limitation of this study was its design. A cross-sectional study did not allow us to clarify the directionality between anxiety and the associated factors. Furthermore, this type of study did not allow us to determine the evolution and follow-up of students’ state of mind. Separately, further studies with the VAS-A and the S-Anxiety scales are needed to check whether the observed proportions and concordances remain unchanged in similar crises, both in clinical and in academic situations, and even study these in health professionals. Finally, the sample size might be an additional limitation; therefore, further studies should assess possible variations in larger samples.

## 5. Conclusions

The innovative nature of this research stands out for its academic context and the results obtained in terms of applicability. The study shows that speedy detection of anxiety using the VAS-A and an in-depth approach with the STAI by teaching staff in crises is possible. Neither scale shows 100% reliability, but according to the scores, all statistical data from the VAS-A scale (error < 5%) are reasonably sound.

## 6. Implications for Practice

The certainty of the ranges of concordance between scales makes the VAS-A feasible for pre-diagnostic anxiety in nursing students. Thus, its initial use is proposed in a simplified form, which is recommended in combination with the S-Anxiety in scores between 5 and 7 in crises (Figure 3). This would significantly reduce the need to administer the STAI, which is longer and more laborious to interpret.

This research also highlights the importance of monitoring anxiety, which may hinder undergraduate nursing students’ effective academic development. This could be done by analyzing and redistributing academic loads and clinical practice, reconsidering and improving teaching methodologies and current assessment systems, and exploring psychological intervention strategies to mitigate the consequences of the pandemic. The overall approach can mitigate the emotional impact of the COVID-19 pandemic. In addition, it is crucial to assess and create resources to help nursing students cope effectively with their emotions during periods of stress, as suggested by Marshall and Wolanskyj-Spinner [36].

## Figures and Tables

**Figure 1 ijerph-19-07053-f001:**
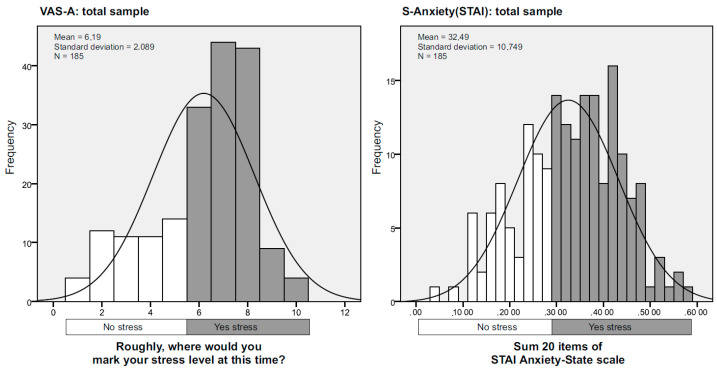
VAS-A and STAI histograms by cut-off point.

**Figure 2 ijerph-19-07053-f002:**
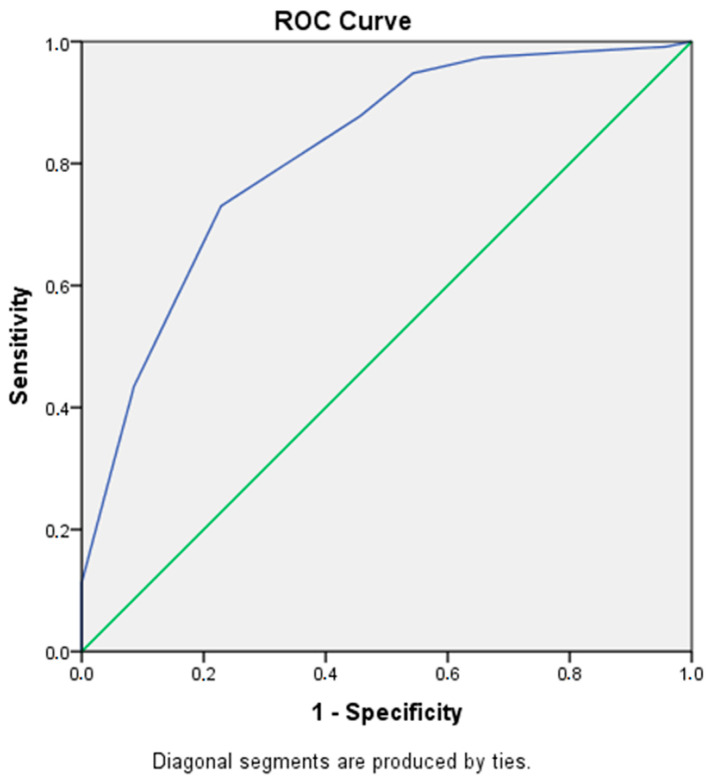
ROC curve.

**Figure 3 ijerph-19-07053-f003:**
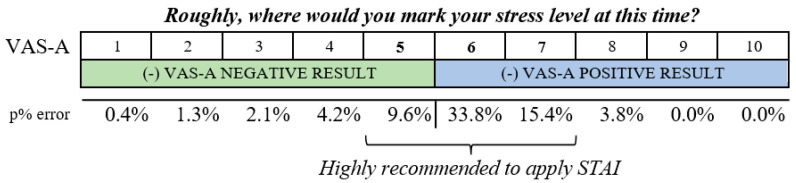
Application of scales according to scores.

**Table 1 ijerph-19-07053-t001:** Characteristics of the sample: number (*n*) and frequencies (%).

Variables		*n*	%
Age *		21.3	4.0
Sex	Men	22	11.9
	Women	163	88.1
Nursing degree year	1st	48	26.0
	2nd	40	21,6
	3rd	45	24.3
	4th	52	28.1
Health worker	No	126	68.1
	Yes	59	31.9
Level of anxiety (VAS) *		6.19	2.09
Level of anxiety (STAI) *		58.43	18.81
S-Anxiety (STAI) *		32.49	10.75
T-Anxiety (STAI) *		25.94	9.97

* Mean and standard deviation (SD).

**Table 2 ijerph-19-07053-t002:** Association between anxiety by VAS-A and S-Anxiety and descriptive variables: number (*n*) and frequency (%).

		VAS-A						S-Anxiety (STAI)					
Factors		No		Yes		*p*	*g*	No		Yes		*p*	*g*
		*n*	%	*n*	%			*n*	%	*n*	%		
Age *		21.9	4.4	21	3.8	0.170	0.23	21.5	3.3	21.4	4.3	0.555	0.09
Sex **	Man	9	17.3	13	9.8	0.155	0.23	11	16.9	11	9.2	0.120	0.12
	Woman	43	82.7	120	90.2			54	83.1	109	90.8		
Nursing degree year **	1st and 2nd	16	30.8	84	63.2	0.000	0.68	28	43.1	72	60	0.027	0.07
	3rd and 4th	36	69.2	49	36.8			37	56.9	48	40		
Health worker **	No	36	69.2	90	67.7	0.838	0.03	44	67.7	82	68.3	0.929	0.02
	Yes	16	30.8	43	32.3			21	32.3	38	31.7		

* Mean and SD. *p*-valor (*t*-student) and effect size (Cohen’s *d*). ** Number (*n*) and frequency (%). *p*-valor (test-Z) and effect size (Hedge’s *g*).

**Table 3 ijerph-19-07053-t003:** Correlation between VAS, S-Anxiety, and T-Anxiety.

	Anxiety (VAS-A)	S-Anxiety (STAI)	T-Anxiety (STAI)
VAS-A	1	0.686 *	0.417 *
S-anxiety (STAI)	0.686 *	1	0.648 *
T-anxiety (STAI)	0.417 *	0.648 *	1

* *p* < 0.001.

**Table 4 ijerph-19-07053-t004:** Cross-tabulation of STAI frequencies: S-Anxiety > 30 and VAS-A > 6.

	S-Anxiety (STAI) > 30				
		No	Yes	*n*	%
VAS-A > 6	No	38	14	52	28.1
	Yes	32	101	133	71.9
	*n*	70	115	185	100
	%	37.8	62.2	100	

**Table 5 ijerph-19-07053-t005:** ROC curve coordinates for sensitivity, specificity, and probability of error.

	VAS-A	Sensitivity	1—Specificity	Probability % of Error	
VAS-A (−) NEGATIVES	1	0.991	0.957	0.4%	of false negatives (1—Sensitivity)
	2	0.983	0.800	1.3%	of false negatives (1—Sensitivity)
	3	0.974	0.657	2.1%	of false negatives (1—Sensitivity)
	4	0.948	0.543	4.2%	of false negatives (1—Sensitivity)
	5	0.878	0.457	9.6%	of false negatives (1—Sensitivity)
VAS-A (+) POSITIVES	6	0.730	0.229	33.8%	of false positives (1—Specificity)
	7	0.435	0.086	15.4%	of false positives (1—Specificity)
	8	0.113	0.000	3.8%	of false positives (1—Specificity)
	9	0.035	0.000	0.0%	of false positives (1—Specificity)
	10	0.000	0.000	0.0%	of false positives (1—Specificity)

## Data Availability

Not applicable.

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
