# Peer review of "Diagnostic Concordance between the Visual Analogue Anxiety Scale (VAS-A) and the State-Trait Anxiety Inventory (STAI) in Nursing Students during the COVID-19 Pandemic"

_ijerph, 2022, doi:10.3390/ijerph19127053_

Round 1

Reviewer 1 Report

The aim of the paper is to establish an effective and efficient anxiety screening device. It would be effective to state this in the title of the article. The argumentation and presentation of data is laid out in an easily followed order. Their assumptions logically follow the evidence revealed by the data collected. We have indeed been through a very difficult and anxiety ridden time in our world history. They accurately draw attention to methodology that is overly long, though accurate. Their tool serves to provide a quick measure that predicts the necessity of follow up for the student.

The assertion that the tool may be of service in redistributing course work for nursing students needs much more in-depth study and consultation with course and curriculum design. Also, the tool does not mitigate the anxiety; it provides an assessment only. Then, leaders can better determine what to do next.

The works cited support the argumentation.

Overall, very interesting and timely article.

Minor revisions recommended regarding claims addressed above.

Reviewer 2 Report

This is a very well-done study, timely, interesting, and topic given COVID-19 and the training of nurses on anxiety measures. The study could expand a bit more on how such a test used proactively can then set up interventions to reduce nursing anxiety to foster a more stable, adaptive, and meaningful nursing team to be better equipped for challenges that may not be easily predicated as the pandemic. This tool could also be used to parse out populations of more emotional stable and less-emotionally stable nurses to be prescriptive of front-line responders of best quality nurses to meet the demands of a pandemic or other eminent medical challenge.

On the writing, never start a sentence with an acronym. 

Regarding the data/statistics all effect sizes should be included. 

Otherwise, this was a really interesting paper and I applaud the authors for their creativity, ingenuity, and effort in updating these tools in the field and hopefully the impact can be global. Job well done!

Reviewer 3 Report

Overview:

The paper addresses a simple question: what is the relationship between anxiety measurement using STAI and VAS-A, and the validity of the results?  It does so within a very specific context of nursing students.  As such it is an important question from a practical viewpoint of monitoring anxiety and responding rapidly and appropriately - a concern of very high importance to the training and retention of nursing skills.

The paper is well documented and appropriately references the key literature on the topic.

Documentation and analysis:

A missing piece of information (unless I missed it!) is when the survey was conducted.  This has relevance to this study because if it is late in the pandemic rather than early, the level and nature of anxiety may be different for different course years. Covid is more than 2 years old so students in years 1-2 have only known the virtual learning approach within the covid environment, whereas senior students have a different experience, including studying through the transitions in teaching modes.  I also wonder whether additional stress and anxiety among the teaching faculty imposed by the changed teaching environment may have spin-off effects on students – this could have a differential effect on different years. I acknowledge that this is not a question central to the theme of comparability of measurement approaches, but could partly explain the unexpected levels of anxiety in years 1 and 2 and years 3 and 4 (lines 216-217).

Table 2: this looks right but just double check that the data in the first two columns is correct.

Limitations and conclusion:

The conclusion is valid and well structured.

Given that this is based on survey results carried out in a very atypical pandemic environment, the limitations of the study are well described. In point of fact, the cited limitations directly point to some of the most interesting aspects that would suggest avenues for further research: especially around the longitudinal evolution an anxiety and synchronic contextual drivers at the micro-scale. The authors should be strongly encouraged to pursue these themes in further research. The current paper provides a very good starting point with a clear indication of an understanding of the gaps in knowledge.

Sample size is a decidedly important consideration since the relationship between anxiety at each year of study is so tightly connected to maturity, experience, confidence and knowledge.  The (primarily) 4th year students who are health workers also will have different anxiety drivers and outcomes that may influence responses, so a longitudinal study that followed individual students across the years of study would add a wealth of information. Another aspect which a larger sample might elicit would be changes in intra-year anxiety (do anxiety levels change across the year for each year of study, allowing for intrusive elements such as exam-anxiety?)

Overall a useful paper and well written. It is thoughtful and adds constructively to the literature on the topic.
